# Pilot Evaluation of Silicone Surrogates for Oral Mucosa Simulation in Craniofacial Surgical Training

**DOI:** 10.3390/biomimetics9080464

**Published:** 2024-08-01

**Authors:** Mitchell D. Cin, Krishna Koka, Justin Darragh, Zahra Nourmohammadi, Usama Hamdan, David A. Zopf

**Affiliations:** 1College of Medicine, Central Michigan University, 1632 Stone St, Saginaw, MI 48602, USA; 2Department of Biomedical Engineering, University of Michigan, Carl A. Gerstacker Building, 2200 Bonisteel Blvd Room 1107, Ann Arbor, MI 48109, USA; kkoka@umich.edu (K.K.); znourmoh@med.umich.edu (Z.N.); 3Department of Molecular and Integrative Physiology, University of Michigan Medical School, 7744 Medical Science II, 1137 Catherine St, Ann Arbor, MI 48109, USA; jwdarrag@umich.edu; 4Department of Otolaryngology-Head and Neck Surgery, University of Michigan Medical School, 1540 E Hospital Dr, Ann Arbor, MI 48109, USA; 5Global Smile Foundation, 106 Access Rd #209, Norwood, MA 02062, USA; uh@gsmile.org

**Keywords:** silicone, mucosa, surgery, simulation, craniofacial, otolaryngology, education

## Abstract

Surgical simulators are crucial in early craniofacial and plastic surgical training, necessitating synthetic materials that accurately replicate tissue properties. Recent critiques of our lab’s currently deployed silicone surrogate have highlighted numerous areas for improvement. To further refine our models, our group’s objective is to find a composition of materials that is closest in fidelity to native oral mucosa during surgical rehearsal by expert craniofacial surgeons. Fifteen platinum silicone-based surrogate samples were constructed with variable hardness and slacker percentages. These samples underwent evaluation of tactile sensation, hardness, needle puncture, cut resistance, suture retention, defect repair, and tensile elasticity. Expert craniofacial surgeon evaluators provided focused qualitative feedback on selected top-performing samples for further assessment and statistical comparisons. An evaluation revealed surrogate characteristics that were satisfactory and exhibited good performance. Sample 977 exhibited the highest performance, and comparison with the original surrogate (sample 810) demonstrated significant improvements in critical areas, emphasizing the efficacy of the refined composition. The study identified a silicone composition that directly addresses the feedback received by our team’s original silicone surrogate. The study underscores the delicate balance between biofidelity and practicality in surgical simulation. The need for ongoing refinement in surrogate materials is evident to optimize training experiences for early surgical learners.

## 1. Introduction

Ensuring the fidelity of synthetic materials used in surgical simulators to accurately replicate tissue handling and repair requirements is crucial for effectively training surgeons [1]. However, the correlation between the choice of tissue surrogates and the training outcomes for trainee surgeons remains poorly understood and requires further exploration. High-fidelity surgical simulators that closely mimic human tissue properties are essential for developing the fine motor skills and tactile sensitivity necessary for performing delicate surgical procedures. Our group’s surgical simulators emphasize low-cost manufacturing combined with the high-fidelity simulation of oral mucosa using silicone. Feedback from recent validation studies conducted by craniofacial surgeons and trainees has highlighted several critiques of our lab’s currently deployed silicone surrogate, specifically regarding its tensile elasticity and suture retention capabilities [2,3]. These critiques underscore the need for materials that not only replicate the mechanical properties of human tissues but also withstand the rigors of surgical training.

The literature indicates that human oral mucosa tissues exhibit site-dependent mechanical properties, which vary significantly depending on their location within the oral cavity [4,5]. For example, the buccal mucosa is thicker and more elastic to endure mechanical stress during mastication, whereas the gingival mucosa is firmer to provide stability around the teeth. This site-specific variation is critical for developing realistic surgical simulators that can provide accurate and effective training. To address this problem, our research aims to rigorously test the mechanical characteristics of various silicone surrogates and qualitatively assess their biofidelity based on subjective evaluations by expert craniofacial surgeons. By doing so, we aim to identify a composition of materials that closely approximates the mechanical properties of human oral mucosa during surgical rehearsal. This study will fill a gap in the literature, as no current studies specifically evaluate the simulation of oral mucosa with silicone materials.

### 1.1. A History of Surgical Education

Surgical education has evolved from its infancy as an informal apprenticeship to the sophisticated and formal structured residency training programs we see today [6]. Despite the formalization of medical education in the late 19th and early 20th centuries, surgical practice remained largely unregulated, lacking standardized guidelines and resulting in varied outcomes until mid-century efforts introduced more rigorous standards and oversight [6]. The establishment of formal residency programs in the 20th century, inspired by the work of William Halsted at Johns Hopkins Hospital, introduced a structured and progressive training system that emphasized hands-on experience, rigorous academic study, and a gradual increase in responsibility [6]. 

A foundational technique in surgical education that emerged during this evolution is the “See One, Do One, Teach One” method [7]. This approach involves a stepwise learning process where a trainee first observes a procedure (See One), then performs the procedure under supervision (Do One), and eventually teaches the procedure to another trainee (Teach One) [7,8]. While this method has been instrumental in surgical training, concerns about patient safety and the need for more comprehensive training have prompted calls for its adaptation and enhancement [7,9]. The purpose of this paper is to demonstrate that the basis of the traditional teaching method is still valid in surgical training and can be combined with other learning adjuncts.

### 1.2. Teaching for Rare Surgical Methods

The Halstedian model, traditionally rooted in the apprenticeship approach to surgical training, underscores the value of experiential learning through direct observation, practice, and subsequent teaching of surgical procedures [7,10,11,12]. This model thrives on the hands-on experience it provides, effectively catering to kinesthetic learners by immersing them in real-world surgical contexts directly within the operating room [7]. The immediate application of observed techniques allows for rapid skill acquisition, which is crucial in the demanding pace of medical environments. Furthermore, the model facilitates a dynamic mentorship environment where experienced surgeons provide direct feedback and guidance, enhancing the professional development of trainees [7].

However, the model also presents significant drawbacks. The variability in case exposure can severely limit a trainee’s experience, particularly with rare or complex conditions, thereby impacting their ability to handle diverse surgical challenges [7,13]. The pressure to perform procedures after minimal exposure not only elevates the risk of errors but also places immense stress on trainees, potentially compromising both patient safety and the trainee’s confidence. Additionally, the inconsistency in learning opportunities—highly dependent on the available cases and the mentor’s teaching capabilities—can result in a patchy educational experience, with trainees becoming proficient in some areas while remaining inexperienced in others [7].

In addressing these limitations, simulation-based training emerges as a complement to the Halstedian model, offering a controlled, risk-free environment where trainees can repetitively practice both common and rare surgical procedures. Simulations ensure a standardized training experience that is not limited by the availability of specific cases and can be regularly updated to reflect the latest surgical techniques [14,15]. Although simulations are invaluable for developing technical skills, they cannot replicate the pressures and unpredictabilities of actual surgical environments. Therefore, an integrated approach that combines the real-life experiences of the Halstedian model with the controlled, repetitive practice opportunities provided by modern simulations may offer the most effective training paradigm, ensuring that surgical residents are thoroughly prepared for a wide range of clinical situations [14,15,16]. The introduction of surgical simulators in all fields of surgical education has demonstrated an increased operative performance when measured using Global Rating Scale (GRS) scores and led to significant reductions in operative time [17].

### 1.3. 3D Modeling for Educational Insight

The adoption of 3D modeling in surgical simulation was largely driven by the limitations associated with animal models [10,18]. This technology, particularly in the fabrication of simulators, offers significant benefits such as slower decomposition rates, enhanced mucosal texture, and, notably, a reduction in the need for animal sacrifice [19]. One striking advancement includes 3D-printed models infused with hydrogels, which provide a highly realistic replication of human tissue structures, offering an anatomical fidelity that is crucial for surgical training [20].

In the field of craniofacial surgery, including Plastic Surgery, Otolaryngology—Head and Neck Surgery, and Oral Maxillofacial surgery, the precision and sophistication of 3D modeling have surpassed traditional methods like cadaveric dissection, providing nasofacial models with excellent anatomical accuracy [21]. The application of 3D modeling extends into preoperative planning as well, where specialized software is utilized to create detailed maps of patient-specific defects or lesions [21,22,23]. This innovation enables surgeons to conduct rehearsals of complex surgical procedures tailored to the unique anatomical challenges of individual patients, enhancing the precision and safety of surgical interventions [21].

### 1.4. Functional 3D Models Enable On-Demand Practice 

Functional 3D surgical models are transformative tools that allow for immediate, real-time feedback during practice, enabling surgeons to correct mistakes as they occur [17,24]. These models are crafted to closely mimic the tactile properties of human tissues, such as elasticity, tensile strength, and resistance, providing a realistic sense of how tissues react under surgical manipulation [24]. This realism is crucial for surgeons to refine the precise motor skills necessary for minimizing tissue damage and improving patient outcomes during delicate surgical procedures [24,25,26].

Moreover, functional 3D models’ versatility extends to their availability for practice anytime [27]. This feature allows for continuous and self-paced skill enhancement, which is not confined to scheduled training sessions [24]. Such accessibility ensures that surgical trainees can consistently practice and improve their techniques, thus maintaining and advancing their skills independently of formal educational opportunities.

## 2. Materials and Methods

Fifteen platinum silicone-based surrogates were constructed, utilizing variable silicone hardness and variable slacker percentages (Table 1). Platinum-cure silicone (Dragon Skin, Smooth-On, Inc., Easton, PA, USA) with hardness of 10A and 00-45 was utilized, with silicone slacker (Smooth-On, Inc., Easton, Easton, PA, USA) percentages between 10% and 30%, based on our laboratory’s historical success in oral maxillofacial tissue simulation and industry standards. The surrogates were designed to include defects ranging from 0.5 cm to 2.0 cm at 0.5 cm increments and a solid exposed 4 × 2 × 2 cm tab for performing suture pull-out and strain testing (Figure 1).

The silicone surrogate models were crafted using molds constructed through fused deposit modeling (FDM) 3D-printing technology, employing polylactic acid (PLA) plastic as the primary material (as depicted in Figure 1). To achieve optimal uniformity and texture, the silicone was degassed under a vacuum of 25–30 inHg for 3 min to remove large bubbles created during the mixing process. Subsequently, the molds were filled with degassed silicone and left to cure at a controlled temperature of 40 °C for 4 h. Silicone is a thermosetting material and must be placed in a higher temperature environment to expedite the curing process and facilitate the cross-linking of silicone polymers. Vacuum degassing, mixing, and curing were all completed per each manufacturer’s recommendations. Each surrogate was uniquely tagged with a randomized three-digit identifier to maintain the anonymity of the material composition during the testing phase.

The framework for testing these surrogates was robust, involving initial evaluations conducted by a dedicated laboratory staff member, followed by a detailed assessment from an expert craniofacial surgeon. The surrogates were securely positioned on testing stands crafted from FDM 3D-printed PLA to ensure stability (also shown in Figure 1). The evaluation process encompassed a variety of material tests which included assessing tactile sensation, evaluating tactile hardness, conducting needle puncture tests, measuring cut resistance against a #10 surgical blade, testing suture retention through manual pull-out tests, examining defect repair capabilities, and determining tensile elasticity. This last measure was evaluated both manually through finger pinch and mechanically by testing suture failure/pull-out. The sutures selected for these tests were 4-0 vicryl sutures equipped with an RB2 needle, typically used for mucosal closure in craniofacial surgeries.

From the array of tests performed, three samples were identified as superior based on their cumulative performance in suture retention, their ability to successfully repair significant defects, and their tensile elasticity as measured using finger pinch and suture failure tests. These samples were designated as 153, 925, and 977 (referenced in Figure 2). After the initial evaluations, these selected samples were further subjected to surgical handling assessments conducted by two additional expert craniofacial surgeons. The assessments were based on both a quantitative 1–5 Likert scale and qualitative free-text feedback, providing a comprehensive view of each surrogate’s performance under simulated surgical conditions. Validation was provided by expert surgeons, with their subjective evaluation serving as a high-fidelity affirmation of these models. 

## 3. Results

While it was found that none of the surrogates fully replicated the complete range of mechanical features characteristic of the native oral mucosa, they exhibited some commendable performance attributes. The study focused on three selected surrogates, all of which demonstrated an ability to perform suture defect repairs at defect sizes of 0.5 cm, 1 cm, 1.5 cm, and 2.0 cm without experiencing failure. Furthermore, these surrogates displayed consistent cutting performance when tested with a number 10 blade. Among the surrogates, sample 977 distinguished itself by achieving the highest scores across multiple evaluation categories. These categories included tactile sensation, tactile hardness evaluation, suture retention to manual pull-out, and tensile elasticity—assessed both via manual finger pinch and suture failure/pull-out tests. The mean performance scores for the surrogate samples were 3.75 for sample 153, 3.88 for sample 925, and an impressive 4.38 for sample 977, on a scale of 5, indicating a significant level of competence in these synthetic models (refer to Table 2).

Evaluation scores of samples 153, 925, and 977 improved markedly compared to the original surrogate model used by the research team, sample 810 (illustrated in Figure 3). This substantial increase underscores the advancements made in the development of these surrogates, highlighting their potential utility in clinical and educational settings where the replication of the mechanical properties of native oral mucosa is required for training or procedural practice.

## 4. Discussion

### 4.1. Current Challenges & Future Directions

In evaluating the effectiveness of silicone-based surgical simulators, several key challenges emerged. One significant issue was balancing high fidelity with practical usability. While the optimized silicone formulation enhanced the realism of the surrogates, expert craniofacial surgeons highlighted numerous areas for improvement in the free-text response boxes. They reported difficulties in material manipulation, noting significant hand strain and prolonged procedural times. These responses underscore the challenge of balancing low-cost manufacturing with high-fidelity anatomic simulation. This feedback from expert evaluators, pioneers in their field with over 30 years of clinical experience, emphasized the nuanced distinctions in biofidelity and the necessary representation of surgical anatomy for training early surgical learners. These findings define limitations inherent to our selected materials and push us to reconsider alternative materials and/or additional silicone additives. Additionally, the commentary on prolonged procedural time influences our consideration of the settings in which our simulators may be deployed. Surface texture and elasticity presented another hurdle. 

Despite efforts to improve these characteristics, replicating the precise microstructure of oral mucosa remains complex. The oral mucosa is a composite structure composed of multiple layers, each with distinct mechanical properties. These layers include the epithelium, lamina propria, and submucosa, which collectively contribute to the unique tactile and elastic characteristics of the tissue. Our single-composition silicone models cannot adequately mimic this layered complexity, resulting in a surrogate that falls short of accurately representing the native tissue. Achieving the right balance between flexibility and firmness is crucial, as it impacts how the tissue reacts to surgical interventions like suturing and cutting. Continuous refinement of material composition and surface treatments is necessary to address these nuances.

Additionally, the variability in the tactile properties of oral mucosa due to factors such as age and disease state further complicates the creation of a universally accurate surrogate. For instance, the oral mucosa of younger individuals typically exhibits greater elasticity and resilience compared to older individuals, whose tissues may be more brittle and less pliable. This increased brittleness in older individuals’ tissues can lead to different responses to surgical manipulation, such as tearing or reduced ability to retain sutures, which are critical factors for training purposes. The elasticity and pliability of younger tissue allow for smoother incisions and suturing, providing a different tactile experience that must be accurately replicated for effective training. Similarly, disease states such as diabetes, cancer, or chronic inflammatory conditions significantly alter the mechanical properties of the oral mucosa. In patients with diabetes, for instance, reduced blood flow and chronic inflammation can result in thicker, more fibrotic tissue, which is less elastic and more difficult to manipulate surgically. This change in tissue quality can affect everything from the ease of incision to the success of suturing and healing processes. Cancerous tissues may be infiltrated with malignant cells, altering their density and elasticity, and making them behave differently under surgical tools compared to healthy tissues. Chronic inflammatory conditions can lead to persistent tissue changes such as fibrosis or the presence of granulomas, which also change the tactile feedback a surgeon receives during procedures.

In addition to challenges with materials, the variability among surgeon evaluations is of great importance. The selected experts for evaluation and assessment were allowed to use their best judgment to analyze the surgical models so as to simulate the way that individual surgeons vary in their practice. We exclusively provided the models and specified the type of procedure. For future studies, developing criteria and standard guides for how surgeons may interact with the models may create more standardization to this methodology, albeit stray further away from emulating the realities of surgical practice. Qualitative evaluations provided by expert surgeons, while invaluable for assessing the surrogates’ performance in relation to their surgical experience, also present a limitation due to the small sample size of the surgeons involved.

The broader field of surgical simulation also faces significant challenges. High development and implementation costs for advanced simulation technologies pose a major barrier [18,28]. Although the long-term benefits of simulation-based training are clear, the initial investment can be prohibitive for many institutions, particularly those in resource-limited settings. This financial barrier restricts the widespread adoption of high-fidelity simulators, which are essential for standardized surgical education globally [2,28]. Moreover, establishing a systematic and ongoing feedback loop with experts is resource-intensive [29,30]. While invaluable, coordinating regular workshops and collaborative sessions across multiple institutions and regions can be logistically challenging [2,28]. Such feedback mechanisms are crucial for gathering practical insights and ensuring simulators evolve to meet the real-world needs of surgical trainees.

Future studies should consider several specific design improvements to address the feedback from expert surgeons. First, optimizing the silicone formulation by adjusting the ratio of silicone 10A and silicone 00-45 and experimenting with different percentages of slacker can help achieve a more balanced material that is easier to manipulate while maintaining high fidelity. Exploring alternative additives or materials that reduce hand strain and improve procedural efficiency could be beneficial. Incorporating advanced manufacturing techniques such as multi-material 3D printing could allow for the creation of more complex, anatomically accurate models with varying mechanical properties [14,30]. This would provide a more realistic training experience by mimicking the heterogeneity of human tissues.

Additionally, formalized quantitative mechanical testing should be conducted to assess the tensile and compressive strength of the materials compared to real oral mucosa. Although our selected series of 15 surrogates was an adequate representation of possible candidates, there is value in evaluating the influence of refined changes in silicone hardness and additives. For example, understanding the performance of surrogate sample 977 versus a new sample with a 1:0.66 ratio of 10A to 00-45 silicone or a sample with 5% or 15% slacker would provide deeper insights into optimizing material properties. Addressing these challenges requires a multifaceted approach, combining advancements in material science, collaborative feedback mechanisms, and strategic investments. Through ongoing innovation and dedicated efforts to refine and improve the models, the goal is to contribute to the global advancement of surgical education and ultimately enhance patient care outcomes.

### 4.2. Biomimicry of Surrogate Materials 

For surgeries involving delicate structures like the oral mucosa, the precise replication of tissue properties ensures that trainees can develop the necessary skills and techniques in a controlled, risk-free environment. When the simulator materials accurately reflect the properties of biological tissues, they provide realistic resistance during suturing, cutting, and manipulation [1,31]. This realism is critical for developing fine motor skills and tactile sensitivity. It also helps trainees build muscle memory and confidence, which are vital for performing actual surgeries with precision and care. 

In plastic and craniofacial surgery, the ability to accurately simulate the feel and response of human tissues is particularly important due to the complex anatomical structures and the critical functions they serve. The oral mucosa, for example, has unique mechanical properties that vary depending on its location within the mouth [32,33]. The buccal mucosa, which lines the inside of the cheeks, is thicker and more elastic to withstand the constant movement and mechanical stress during mastication [32,33]. In contrast, the gingival mucosa around the teeth is firmer and less elastic, providing a stable environment for the teeth and resisting the forces applied during chewing [32]. Additionally, the sublingual mucosa under the tongue is highly flexible and delicate, accommodating the movements of the tongue [33]. These properties affect how the tissue reacts to surgical interventions, such as incisions, suturing, and handling. If a simulator fails to replicate these nuances, trainees may develop improper techniques or fail to gain the necessary confidence and proficiency required for successful surgeries. 

Moreover, high-fidelity simulation models help hone surgeons’ skills in a safe and controlled environment, reducing the risk of complications during actual surgeries. The tactile realism provided by these simulators ensures that the muscle memory and hand-eye coordination developed during training are directly applicable in real clinical settings [15,34]. This is especially critical in procedures involving delicate tissue manipulation, where the margin for error is minimal. 

Future research should consider specific design improvements based on the feedback from expert surgeons. Optimizing the silicone formulation by adjusting the ratio of silicone 10A and silicone 00-45 and experimenting with different percentages of slacker can help achieve a more balanced material that is easier to manipulate while maintaining high fidelity. Exploring alternative additives or materials to reduce hand strain and improve procedural efficiency would be beneficial. Incorporating softening agents like silicone oil or plasticizers can adjust the silicone surrogates’ flexibility and ease of manipulation [18]. Additionally, using biocompatible hydrogels, which exhibit properties similar to soft tissues, could provide a more lifelike tactile experience while reducing the physical effort required during procedures [18]. Integrating elastomeric polymers such as thermoplastic elastomers (TPEs) can also offer enhanced flexibility and durability, making the surrogates more responsive to handling [35]. Furthermore, advanced composites that blend silicone with other materials like polyurethane or polyethylene glycol (PEG) could be explored to achieve the desired mechanical properties. These materials are known for their elasticity and strength, which could help create realistic and easier-to-work-with surrogates [35]. Utilizing materials with self-lubricating properties or surface treatments that reduce friction can also improve procedural efficiency by making tissue manipulation smoother and less strenuous [18,35].

Utilizing advanced manufacturing techniques like multi-material 3D printing enhances surgical simulators by closely replicating human tissue heterogeneity [36,37]. This technology allows for the creation of models with distinct mechanical properties in a single print cycle, where different materials are strategically placed to mimic tissue-specific characteristics [35]. For instance, softer materials simulate flexible tissues like sublingual mucosa, while tougher materials reproduce the durability of gingival mucosa. This approach not only increases the realism of surgical training models but also broadens the training scope by allowing trainees to experience varied tissue responses during procedures [35]. The integration of diverse materials into one model also prepares surgeons for complex clinical scenarios, enhancing both tactile feedback and procedural readiness. Through multi-material 3D printing, surgical education advances significantly, offering more accurate and comprehensive training tools.

Enhancing the surface texture and elasticity of the surrogates to replicate the tactile feedback of human tissues better can significantly improve the training experience by providing more realistic haptic feedback during surgical procedures. Achieving this level of realism can be accomplished by using surface treatments or coatings that mimic the microstructure of oral mucosa. Establishing a continuous feedback loop with expert surgeons through regular testing and evaluation will ensure that the surrogates are iteratively improved based on practical, real-world usage. Regular workshops and collaborative sessions with surgeons can provide ongoing insights into the surrogates’ performance and highlight improvement areas. 

### 4.3. Local Production, Customization, and Training for Enhanced Accessibility

Exploring the local production and customization of simulators presents a promising avenue for enhancing accessibility and reducing costs. By establishing localized manufacturing facilities, institutions can tailor the production of simulators to meet specific regional needs, thus reducing the dependency on imported models and associated logistical expenses. Customization allows for the development of simulators that cater to the unique anatomical and procedural training requirements of diverse populations, further improving the relevance and effectiveness of the training tools.

To achieve this, there is a critical need for comprehensive training and capacity-building for educators and technicians in the use of simulators. Educators must be adept at integrating these tools into their curricula, while technicians require specialized skills to maintain and optimize the functionality of the simulators. Implementing targeted training programs will ensure that both educators and technicians are well-equipped to utilize these advanced tools effectively, thereby maximizing their educational potential.

### 4.4. Anticipated Challenges and Strategies for Refinement

The refinement of silicone surrogates, while promising, poses several anticipated challenges. Achieving an optimal balance between biofidelity and practical usability remains complex. The nuanced mechanical properties of native oral mucosa, including its site-dependent variations, are difficult to replicate fully with silicone alone. Continuous experimentation with silicone formulations and the incorporation of advanced materials and additives will be necessary to address these challenges. Specifically, refining the ratios of silicone 10A and silicone 00-45, and exploring the inclusion of softening agents, could enhance the flexibility and handling characteristics of the surrogates.

Collaborative efforts with other institutions and industries will be pivotal in advancing the development of these training tools. Partnerships with material scientists, biomedical engineers, and clinical experts can facilitate the integration of cutting-edge technologies and innovative materials into the design of the surrogates. Additionally, collaboration with the medical device industry could provide access to specialized manufacturing techniques and quality control processes, ensuring the production of high-fidelity simulators that meet rigorous standards.

### 4.5. Accessible & Equitable Practice 

The impact of surgical simulators can be transformative in resource-limited settings. These environments often face significant challenges, such as limited access to live patients for practice, a scarcity of cadaveric specimens, and inadequate training facilities [38]. High-fidelity surgical simulators address these challenges by providing an accessible and practical alternative to surgical education. By replicating the mechanical properties of human tissues, these simulators allow trainees to practice complex procedures repeatedly, enhancing their skills and confidence without the ethical and logistical issues associated with using live patients or cadavers [39].

The cost-effectiveness of these simulators is particularly advantageous in resource-limited settings. Surgical training with cadaveric specimens is not only expensive but also logistically challenging due to the need for specialized storage and handling facilities [38]. On the other hand, surgical simulators can be produced at a lower cost and used multiple times, making them a more sustainable option for continuous training. Surgical simulators can be easily transported and set up in various locations, including rural or underserved areas where access to medical training facilities is limited. This portability ensures that high-quality surgical education is not confined to urban centers or well-funded institutions but can reach a broader audience of medical students and residents. As a result, healthcare professionals in these settings can acquire the necessary skills and competencies to perform complex surgeries, ultimately improving patient outcomes and addressing healthcare disparities. Additionally, the use of surgical simulators in resource-limited settings can facilitate more standardized training, ensuring that all trainees receive a consistent and high-quality education regardless of their geographic location. This standardization is crucial for maintaining high standards of surgical practice and enhancing the overall quality of healthcare delivery. 

### 4.6. Global Implications in Surgical Education

The advancement of surgical simulation technology holds significant global implications for surgical education, particularly in addressing ethical, practical, and accessibility challenges. Surgical simulation offers a risk-free environment for trainees to develop and refine their skills without the need for live patients or cadavers [40,41]. In some countries, pig models are commonly used for surgical training due to their anatomical similarities to humans [40,41]. However, in many Muslim-majority areas, the use of pig models is considered haram, or forbidden, due to religious beliefs. This cultural difference presents a challenge in standardizing surgical training across diverse regions. High-quality silicone simulators offer a viable solution, allowing for realistic, ethical, and culturally sensitive training that can be standardized globally. By providing an alternative to animal models, silicone simulators ensure that all trainees can receive comprehensive, hands-on experiences. 

Simulation-based training enhances continuing education and skill maintenance for practicing surgeons worldwide. In clinical practice, surgeons may not frequently encounter rare or complex procedures, making it difficult to maintain proficiency [40,41]. Surgical simulators provide a platform for ongoing education, allowing surgeons to regularly practice these challenging procedures and stay updated with the latest techniques and advancements. This continuous professional development is vital for maintaining high standards of patient care and surgical outcomes, irrespective of the practitioners’ geographic location.

### 4.7. Impact on Patient Care Outcomes

The ultimate goal of this research is to improve patient care outcomes through enhanced surgical training. By providing high-fidelity simulators that accurately replicate the mechanical properties of human tissues, trainees can develop the fine motor skills and tactile sensitivity necessary for performing delicate surgical procedures with precision and confidence. The improvements in training facilitated by these advanced simulators are expected to translate into better surgical performance, reduced operative times, and lower complication rates, thereby directly benefiting patient care.

Moreover, the widespread adoption of these simulators in resource-limited settings can address significant disparities in surgical education. By making high-quality training tools accessible to a broader range of trainees, irrespective of their geographic location or institutional resources, the research aims to contribute to a more equitable distribution of surgical expertise. This, in turn, will enhance the overall quality of healthcare delivery and patient outcomes globally.

## 5. Conclusions

Through rigorous testing and expert evaluation, we have identified a specific silicone composition, particularly in surrogate sample 977, that offers superior performance in tensile elasticity and suture retention capabilities. This sample demonstrated a closer approximation to the mechanical properties of natural oral mucosa than our initial models, suggesting a promising direction for future development. The findings from this study highlight the intricate balance required between achieving high biofidelity and maintaining practicality in surgical training tools. While the selected silicone surrogate showed improved physical characteristics conducive to surgical training, feedback from expert evaluators also indicated areas for further enhancement, especially concerning the ease of material manipulation and procedural efficiency. This feedback is invaluable as it illustrates the real-world challenges surgeons encounter, guiding our next steps in materials research and simulator design.

Moving forward, ongoing refinement and innovation in the composition and testing of synthetic tissues are essential. Expanding the range of mechanical testing and exploring a broader array of silicone formulations will be crucial in honing the biofidelity of our surrogates. These advancements are anticipated to significantly contribute to the field of surgical simulation by providing more accurate and reliable training tools. The enhanced fidelity of these surrogates will better prepare surgical trainees for real-world procedures, thereby improving their skills and confidence. 

Moreover, the potential implications for surgical education are profound. High-fidelity simulators that closely mimic the mechanical properties of human tissues will allow for more effective and realistic training experiences. This, in turn, will facilitate the development of fine motor skills and tactile sensitivity necessary for performing delicate surgical procedures with precision. By addressing the identified limitations and incorporating ongoing feedback from expert evaluators, we aim to develop training tools that not only mimic the physical attributes of human tissues but also enhance the learning curve of nascent surgeons.

Ultimately, our goal remains steadfast—to improve patient outcomes in the clinical setting. The enhanced training provided by these advanced simulators is expected to translate into better surgical training and early performance, reduced operative times secondary to more rapidly attained surgical proficiency and lower complication rates. This research underscores the importance of continuous innovation in surgical simulation to meet the evolving needs of surgical education and improve healthcare delivery worldwide.

## Figures and Tables

**Figure 1 biomimetics-09-00464-f001:**
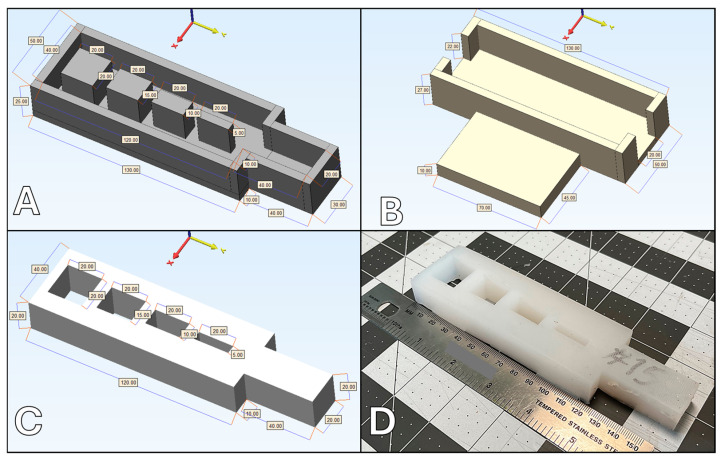
Silicone surrogate testing components: (**A**) surrogate mold; (**B**) surrogate testing stand; (**C**) computer-aided design (CAD) render of resultant silicone-based surrogate sample; (**D**) image of resultant silicone-based surrogate sample. All measurement citations are in millimeters (mm).

**Figure 2 biomimetics-09-00464-f002:**
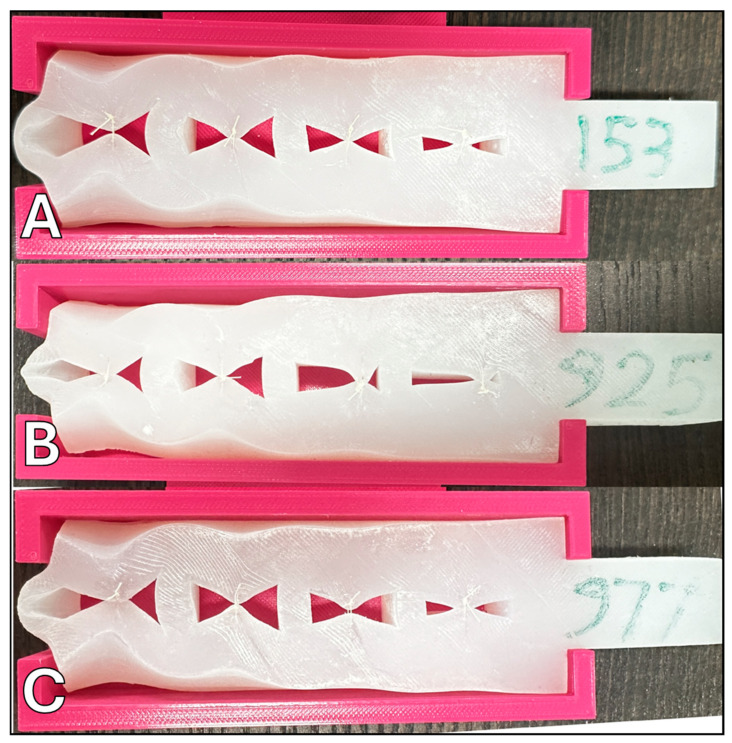
Top-performing surrogate samples following surgical handling evaluation by expert surgeons: (**A**) surrogate sample 153; (**B**) surrogate sample 925; (**C**) surrogate sample 977.

**Figure 3 biomimetics-09-00464-f003:**
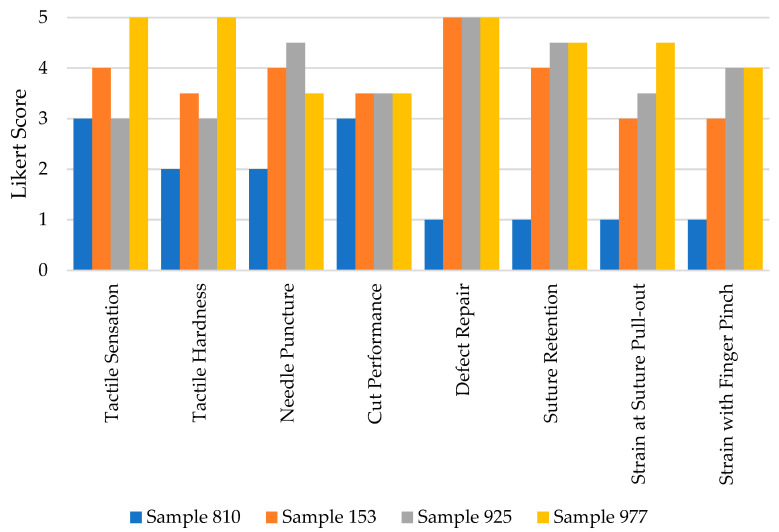
Evaluation scores of our original and top-performing surrogate samples.

**Table 1 biomimetics-09-00464-t001:** Silicone-based surrogate sample candidate population.

Sample #	Silicone 10A	Silicone 00-45	Silicone Additive
810	100 g	0 g	10 g (10%)
456	100 g	0 g	20 g (20%)
925	100 g	0 g	30 g (30%)
926	75 g	25 g	10 g (10%)
153	75 g	25 g	20 g (20%)
370	75 g	25 g	30 g (30%)
977	50 g	50 g	10 g (10%)
445	50 g	50 g	20 g (20%)
642	50 g	50 g	30 g (30%)
966	25 g	75 g	10 g (10%)
876	25 g	75 g	20 g (20%)
864	25 g	75 g	30 g (30%)
643	0 g	100 g	10 g (10%)
498	0 g	100 g	20 g (20%)
745	0 g	100 g	30 g (30%)

**Table 2 biomimetics-09-00464-t002:** Expert craniofacial surgeon evaluation domain scores of the silicone surrogates.

**Domain**	**Sample 153**	**Sample 925**	**Sample 977**
Tactile Sensation	4.0	3.0	5.0
Tactile Hardness	3.5	3.0	5.0
Needle Puncture Performance	4.0	4.5	3.5
Cut Performance	3.5	3.5	3.5
Defect Repair Performance	5.0	5.0	5.0
Suture Retention to Manual Pull-out	4.0	4.5	4.5
Strain at Suture Pull-out	3.0	3.5	4.5
Strain with Finger Pinch	3.0	4.0	4.0
Average	3.75	3.88	4.38

## Data Availability

All data generated or analyzed during this study are included in this published manuscript.

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
