# Peer review of "Pilot Evaluation of Silicone Surrogates for Oral Mucosa Simulation in Craniofacial Surgical Training"

_biomimetics, 2024, doi:10.3390/biomimetics9080464_

Round 1

Reviewer 1 Report

Comments and Suggestions for Authors

Dear Authors,
I have reviewed your manuscript with great interest. I appreciate the effort put into this study and the potential implications of your findings.

Overall:
The main question addressed by the research is how to find a composition of materials for surgical simulators that closely replicates the properties of native oral mucosa during surgical rehearsal by expert craniofacial surgeons. The use of platinum silicone-based surrogate samples for surgical simulation is relevant. The focus on tactile sensation, hardness, needle puncture, cut resistance, suture retention, defect repair, and tensile elasticity evaluation is original and important. The study aims to improve upon existing silicone surrogates used in surgical training and adds to the subject area by identifying a refined silicone composition that directly addresses feedback received from the team’s original silicone surrogate. It emphasizes the balance between biofidelity and practicality in surgical simulation. The authors should consider providing more details about the specific composition of the platinum silicone-based surrogate samples. Further controls could involve comparing the surrogate performance with actual oral mucosa to validate the fidelity. The conclusion that sample 977 exhibited the highest performance should be supported by specific evidence from the evaluation. The study should clarify how the refined composition addresses the specific areas of improvement highlighted in the critiques of the original surrogate.

More in detail:
Introduction
The introduction starts with a reference to critiques of the silicone surrogate, but it could benefit from a clearer statement of the research problem. Consider rephrasing sentences for better readability. For example, “Literature indicates that human oral mucosa tissues exhibit site-dependent mechanical properties…” could be simplified. The introduction mentions the need for materials that replicate human tissue properties but lacks specific context. What are the challenges faced by current surgical simulators? Why is fidelity important? Explicitly state the gap in the field that this study aims to address. For instance, highlight the lack of studies evaluating oral mucosa simulation with silicone materials. Consider emphasizing the novelty of this research by comparing it to existing literature. The transition from discussing silicone surrogates to surgical education history could be smoother. Consider providing a more direct link or a brief summary of the gap before delving into historical context. The historical context is well-presented, but the purpose statement (“demonstrate that the basis of the traditional teaching method is still valid…”) could be more explicit. What will this demonstration achieve? The discussion on the Halstedian model is informative, but it lacks a clear connection to the main research topic (silicone surrogates). Consider integrating the discussion of rare surgical methods more seamlessly into the overall narrative. The section on 3D modeling is well-structured and informative. However, it could benefit from a stronger link to the study’s objective. How does 3D modeling relate to refining silicone surrogates?

Materials and Methods
The use of platinum-cure silicone with varying hardness and slacker percentages is well-documented. The process using FDM 3D-printing technology and PLA plastic is adequately explained. The degassing and curing details are precise; however, it would be beneficial to explain why these specific conditions were chosen. The use of a randomized identifier for each surrogate is a good practice to maintain objectivity. A comprehensive range of tests is described, which is excellent for assessing the surrogates’ performance. It might be helpful to include more details on the standardization of the evaluation process and how the tests mimic real surgical conditions. The criteria for selecting superior samples based on suture retention, defect repair, and tensile elasticity are clear. The use of both quantitative and qualitative feedback from additional expert surgeons is a strong aspect of the methodology. Provide more context on the choice of materials and methods, including any precedent studies or literature that guided these choices. Clarify the rationale behind the specific testing conditions, such as temperature and vacuum pressure.

Results
Sample 977 stands out with the highest scores in tactile sensation, hardness evaluation, suture retention, and tensile elasticity. The comparison with the original surrogate model, Sample 810, shows marked improvement in the new samples. Provide a more detailed analysis of why the surrogates did not fully replicate the mechanical features of native oral mucosa. Include statistical analysis to support the performance scores and the significance of the improvements. Discuss the limitations of the results and how they might impact the utility of the surrogates in clinical and educational settings. Consider discussing the potential for further development based on the results obtained.

Discussion
The manuscript should discuss potential strategies for achieving the desired balance between flexibility and firmness. It would be helpful to include a more detailed plan for future studies, including potential timelines and milestones. Discuss the feasibility of incorporating alternative materials and advanced manufacturing techniques. Provide a clearer roadmap for future research, including how the feedback will be integrated into the design process. Address the limitations of the current study and how they might influence the interpretation of the results. Consider the ethical implications of using alternative materials and the impact on sustainability. The discussion effectively highlights the importance of accurately replicating tissue properties for surgical training. It emphasizes the critical role of high-fidelity simulation models in developing surgeons’ skills. It could benefit from discussing specific examples of how these challenges have been addressed in the past. Establishing a feedback loop with expert surgeons for iterative improvement is a valuable recommendation. Provide more detailed explanations of how the proposed materials and techniques will overcome the current limitations. Discuss the feasibility and cost-effectiveness of implementing the suggested improvements. Consider the ethical and environmental implications of using new materials. Address the scalability of the proposed manufacturing techniques for widespread adoption. Provide more detailed data or case studies to support the cost-effectiveness claims. Discuss potential challenges in implementing simulator-based training in resource-limited settings, such as maintenance and technical support. Explore the possibility of local production and customization of simulators to further enhance accessibility and reduce costs. Address the need for training and capacity-building for educators and technicians in the use of simulators. Discuss the anticipated challenges in further refining the silicone surrogates and how the research team plans to overcome them. Consider mentioning the collaborative efforts with other institutions or industries that might be necessary for advancing the development of these training tools. Reflect on the potential impact of these advancements on patient care outcomes, which is the ultimate goal of the research.

Reviewer 2 Report

Comments and Suggestions for Authors

Dear authors,

Thank you for submitting your manuscript, "Evaluation of Silicone Surrogates for Oral Mucosa Simulation in Otolaryngology-Head and Neck Surgical Training," for consideration in Biomimetics. After a thorough review, I have identified several areas that require improvement before the manuscript can be considered for publication.

Quality of the text:

The manuscript is generally well-written, and the English language is of good quality. However, to improve readability, a few instances of grammatical errors and awkward phrasing should be addressed.

Images:

The quality of the images, particularly Figure 3, could be improved for better clarity.

Methods and description of results:

The methods section should include a detailed description of the materials and techniques used in the study. The sample size of 15 surrogates may be considered small and not "an adequate representation". The rationale for selecting this sample size should be provided. The results section adequately describes the findings, but the authors should consider including more quantitative data to support their conclusions.

Aim and conclusion:

The study's aim is clearly stated in the introduction, and the conclusion addresses the main findings. However, the authors should elaborate on how their results contribute to the field of surgical simulation and the potential implications for surgical education.

Limitations:

The authors have acknowledged some limitations of their study, such as the challenge of balancing high fidelity with practical usability. However, they should also discuss the potential limitations of using a small sample size and the need for further validation studies with a larger cohort of expert surgeons.

Weak points:

1. Small sample size of silicone surrogates tested

2. Limited quantitative data to support conclusions

3. Lack of discussion on the broader implications of the findings for surgical education

Weakest point:

The weakest point of the manuscript is the small sample size of silicone surrogates tested. The authors should provide a clear justification for the chosen sample size and discuss how this may impact the generalizability of their findings.

The authors should consider renaming the research as a PILOT STUDY.

Comments on the Quality of English Language

Minor editing of the English language is required

Round 2

Reviewer 1 Report

Comments and Suggestions for Authors

Dear authors,

I have carefully reviewed your revised manuscript and I am pleased to inform you that I find it acceptable for publication. You have addressed all of the issues that I raised in my previous comments and improved the quality and clarity of your work. I appreciate your efforts and cooperation in this process.

Best regards

Author Response

Dear reviewer, thanks for your review

Reviewer 2 Report

Comments and Suggestions for Authors

Based on my review of the manuscript, I believe it is close to being ready for publication, but there are a few areas that could benefit from minor revisions:

Discussion: The discussion is comprehensive, but it could be strengthened by more explicitly addressing the limitations of the current study and providing more specific suggestions for future research directions.

Language: Overall, the writing is clear and appropriate for a scientific publication. However, a final proofreading pass could help catch any minor grammatical or stylistic issues.

Comments on the Quality of English Language

Overall, the writing is clear and appropriate for a scientific publication. However, a final proofreading pass could help catch any minor grammatical or stylistic issues.
